# Catalytic 1,1-diazidation of alkenes

Wangzhen Qiu[1,2], Lihao Liao [1,2] ✉, Xinghua Xu[1], Hongtai Huang [1], Yang Xu[1] & Xiaodan Zhao [1] ✉

Compared to well-developed catalytic 1,2-diazidation of alkenes to produce vicinal diazides, the corresponding catalytic 1,1-diazidation of alkenes to yield geminal diazides has not been realized. Here we report an efficient approach for catalytic 1,1-diazidation of alkenes by redox-active selenium catalysis. Under mild conditions, electron-rich aryl alkenes with *Z* or *E* or *Z/E* mixed configuration can undergo migratory 1,1-diazidation to give a series of functionalized monoalkyl or dialkyl geminal diazides that are difficult to access by other methods. The method is also effective for the construction of poly-diazides. The formed diazides are relatively safe by TGA-DSC analysis and impact sensitivity tests, and can be easily converted into various valuable molecules. In addition, interesting reactivity that geminal diazides give valuable molecules via the geminal diazidomethyl moiety as a formal leaving group in the presence of Lewis acid is disclosed. Mechanistic studies revealed that a selenenylation-deselenenylation followed by 1,2-aryl migration process is involved in the reactions, which provides a basis for the design of new reactions.

Alkenes are readily available and abundant feedstock starting materials. They are considered as one of the most popular modules in synthetic chemistry and industrial production. In particular, they can undergo difunctionalization to rapidly form complex molecules by simultaneously incorporating two functional groups[1–4]. For instance, azidative difunctionalization of alkenes is an important way to produce alkyl azides, which are of great importance in synthetic, material, and bioorthogonal chemistry[5–9]. Specially, alkenes can produce vicinal diazides by the introduction of two azido groups via a 1,2-diazidation process (Fig. 1a, left top). In this regard, the conventional methods mainly depend on Pd, Fe, and Cu-catalyzed radical diazidation, in which stoichiometric oxidants such as hypervalent iodines and organic peroxides are required (Fig. 1a, left bottom)[10–16]. As a mild alternative, electrochemical methods using manganese or aminoxyl or copper electrocatalyst have provided an attractive strategy for alkene 1,2-diazidation in recent years (Fig. 1a, middle bottom)[17–20]. Very recently, photochemical 1,2-diazidation of alkene have also been developed via iron-mediated ligand-to-metal charge transfer (LMCT) strategy (Fig. 1a, right bottom)[21,22]. Owing to the unique reactive nature of the π bond, it was difficult to realize another type of diazidation, the alkene

1,1-diazidation (Fig. 1a, right top). Until now, no efficient methods have been demonstrated with respect to this transformation[23], although the obtained geminal diazides[24–26] can serve as energetic molecules[27–29], their cycloaddition derivatives such as bistriazoles are biologically active molecules and ligands[30–32], and they can also act as good precursors for the synthesis of valuable complicated molecules such as nitriles, amides, tetrazoles, pyridines, pyrazines, isoxazoles, and 1,3, 4-oxadiazoles[24–26]. Thus, developing an efficient approach for direct 1,1-diazidation of alkenes is highly desirable for the synthesis of geminal diazide derivatives.

Iodine (III)-mediated migratory 1,1-difluorination of alkenes has attract considerable attention over past decade since it could generate valuable geminal difluorides as products[33–36]. Generally, in this transformation, alkenes go through a regioselective fluorination to form alkyl fluoride intermediates, followed by 1,2-migration of an intramolecular group, and the formed intermediate was trapped by the second fluoride ion[37–42]. Inspired by this reaction in which double nucleophiles are involved as well as group migration, we questioned whether geminal diazides might be constructed in a migratory 1,1-difunctionalization fashion (Fig. 1b). We proposed

[1]Institute of Organic Chemistry and MOE Key Laboratory of Bioinorganic and Synthetic Chemistry, School of Chemistry, IGCME, Sun Yat-Sen University, Guangzhou 510006, P. R. China. [2]These authors contributed equally: Wangzhen Qiu, Lihao Liao. ✉e-mail: liaolh5@mail.sysu.edu.cn; zhaoxd3@mail.sysu.edu.cn

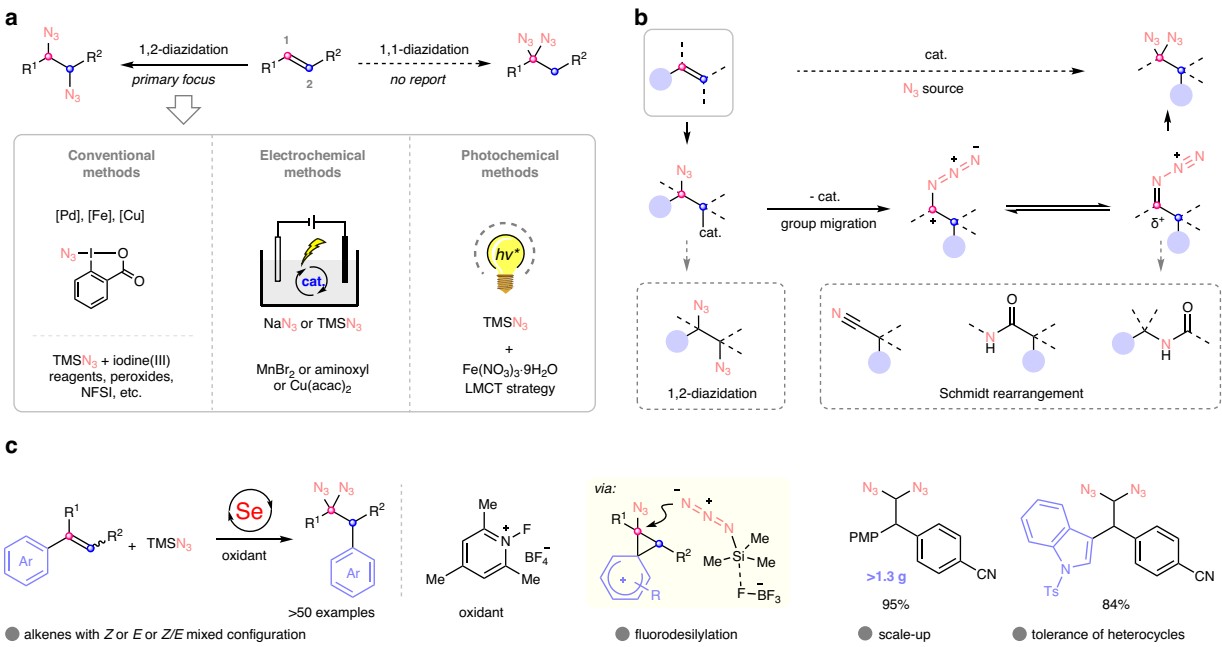

**Fig. 1 | Catalytic diazidation of alkenes. a** Comparison between 1,2-diazidation and 1,1-diazidation of alkenes. **b** Proposed catalytic 1,1-diazidation of alkenes and its challenge. **c** This work: 1,1-diazidation of alkenes enabled by organoselenium catalysis. PMP *p*-methoxyphenyl.

that catalyst-assisted regioselective azidation of alkenes occurred first, followed by intramolecular group migration to generate azide-stabilized intermediate, and then the intermediate underwent a second azidation to give the final products (Fig. 1b, up). This proposed reaction might greatly enrich the field of diazidation of alkenes although it is very challenging owning to the known competitive side reactions such as 1,2-diazidation[10–22] as well as the Schmidt and related rearrangements[43–45] driven by the favorable loss of nitrogen gas from the azido group thermodynamically (Fig. 1b, down). It is worth mentioning that although elegant 1,1-difluorination and 1,1-diborylation of alkenes has been achieved by using iodine catalysts and metal catalysts respectively[37–42,46–48], the azidative version has not be realized yet.

Selenium redox catalysis has emerged as a powerful tool for the functionalization of alkenes[49,50]. This catalysis has enabled the allylic functionalization, direct C−H functionalization, and 1,2-difunctionalization of alkenes[49–53]. In these studies, the involved electrophilic selenium species exhibited the unique properties of carbophilicity toward alkene substrates and the convenience of release from the intermediates, which is different with other electrophilic catalysts. This might offer a possibility for the aforementioned 1,1-difunctionative azidation via rearrangement although selenium catalysis remains elusive for the 1,1-difunctionalization of alkenes.

In this work, we report our discovery that 1,1-diazidation of aryl alkenes with *Z* or *E* or *Z*/*E* mixed configuration can be realized by selenium redox catalysis in the presence of azidotrimethylsilane as azide source and *N*-fluoropyridinium fluoborate as oxidant (Fig. 1c). In the reactions, fluorodesilylation is an important step to accelerate the release of the azido group from azidotrimethylsilane. The reactions not only can tolerate the heterocycle-substituted alkenes, but also can be applied to the construction of polydiazides. TGA-DSC analysis and impact sensitivity tests reveal that these products are relatively safe. Importantly, the reactions can be easily scaled up, and the obtained products can be converted into other valuable molecules expediently. Interesting reactivity of geminal diazides is disclosed as well. This work represents an interesting example of catalytic 1,1-diazidation of alkenes. It also exhibits the unique advantage of selenium redox catalysis, which offers a basis for the design of new reactions.

## Results and discussion

### Reaction design and optimization

We commenced our study with *cis*-stilbene derivative *Z*-**1a** containing *p*-methoxyphenyl group as the model substrate since not only it could be easily prepared via Wittig olefination from the corresponding aldehyde and the phosphonium salt, but also the electron-rich aryl group on the substrate might facilitate the aryl migration which might be crucial for 1,1-difunctionalization (Table 1). Firstly, the reaction was carried out in the presence of 10 mol% of diphenyl diselenide **C1** as the precursor of redox-active selenium catalyst in dichloromethane at room temperature. Sterically hindered *N*-fluoropyridinium fluoborate **O1** was utilized as the oxidant to avoid the undesired amination because the oxidant could serve as a potential endogenous nitrogen nucleophile[54,55]. Azidotrimethylsilane was used as azido source because the silicon cation might consume the fluoride from the oxidant to avoid the undesired fluorination[56]. To our delight, the desired geminal diazide **2a** was generated in 88% yield via 1,2-aryl migration[57–59], while vicinal diazide **3a** was formed in only 4% yield, and the by-products from Schmidt rearrangement were not observed (entry 1). To achieve better results, different catalyst precursors of dichalcogenides were evaluated, and none of them gave better results than **C1** (entries 2–6). It was found that the reactions did not work at all when disulfide and ditelluride were utilized as catalyst precursors. For the former, it might be ascribed to the difficulty for the generation of electrophilic sulfur catalyst since thiolated compounds could be oxidized by N-F reagents to produce relatively stable sulfoxides or sulfones[60]. For the latter, the generated high valent tellurium species was relatively robust and might be unreactive toward alkenes[61]. Next, other oxidants were tested for the reactions. Less bulky fluoropyridinium tetrafluoroborate **O2** led to only moderate yield (entry 7), while other N-F reagents gave nearly no desired geminal diazides along with the formation of vicinal diazides in low yields (entries 8–13). In these reactions, it was found that the tetrafluoroborate anions on the oxidants might be important for this transformation since the oxidants bearing the triflate as anion could not promote the reaction almost (**O1** vs **O3**, **O2** vs **O4**). This phenomenon might be ascribed to that the tetrafluoroborate anion from the oxidants might facilitate the release of azide anions from azidotrimethylsilane via fluorodesilylation[62,63]. In contrast,

## Table 1 | Condition evaluation

| Entry[a] | Catalyst | Oxidant | yield of 2a (%)[b] | yield of 3a (%)[b] | Entry[a] | Catalyst | Oxidant | yield of 2a (%)[b] | yield of 3a (%)[b] |
|---|---|---|---|---|---|---|---|---|---|
| 1 | C1 | O1 | 88 | 4 | 12 | C1 | O7 | 0 | 17 |
| 2 | C2 | O1 | 81 | 11 | 13 | C1 | O8 | 0 | 55 |
| 3 | C3 | O1 | 39 | 7 | 14[c] | C1 | O1 | 90 | 5 |
| 4 | C4 | O1 | 0 | 0 | 15[d] | C1 | O1 | 77 | 18 |
| 5 | C5 | O1 | 0 | 9 | 16 | - | O1 | 0 | 0 |
| 6 | C6 | O1 | 8 | 0 | 17[c,e] | C1 | O1 | 80 | 20 |
| 7 | C1 | O2 | 40 | 9 | 18[c,f] | C1 | O1 | 0 | 0 |
| 8 | C1 | O3 | 0 | 10 | 19[c,g] | C1 | O1 | 0 | 0 |
| 9 | C1 | O4 | 0 | 22 | 20[c,h] | C1 | O1 | 89 | 5 |
| 10 | C1 | O5 | 0 | 0 | 21[c,i] | C1 | O1 | 92 | 8 |
| 11 | C1 | O6 | 0 | 5 | 22[c,j] | C1 | O1 | 96 | 4 |

PMP p-methoxyphenyl, PCP p-cyanophenyl, Ch chalcogen atom.

[a]Conditions: Z-1a (0.05 mmol), azidotrimethylsilane (3.0 equiv), oxidant (2.0 equiv), catalyst (10 mol%), dichloromethane (350 µL), $N_2$, room temperature, 24 h.

[b]Refers to NMR yield with 1,1,2,2-tetrachloroethane as the internal standard.

[c]C1 (5 mol%).

[d]C1 (2 mol%).

[e]1,2-Dichloroethane as solvent.

[f]Chloroform as solvent.

[g]Tetrahydrofuran as solvent.

[h]Acetonitrile as solvent.

[i]E-1a was utilized as substrate.

[j]Dichloromethane (175 µL).

when different type of oxidants such as [bis(trifluoroacetoxy)iodo]-benzene (O8) was utilized instead of O1, only vicinal diazide 3a was formed in moderate yield (entry 13). This result suggested the importance of the oxidative system. Using 5 mol% of C1, 2a was generated in slightly better yield of 90% (entry 14). Further reducing the catalyst loading resulted in the decrease of the product to 77% along with the increase of vicinal difunctionalization product (entry15). No desired product was formed in the absence of any catalyst precursor (entry 16). Furthermore, other common solvents were evaluated (entries 17–20). Acetonitrile was found as nearly effective as dichloromethane, while chloroform and tetrahydrofuran led to no product. When trans-stilbene E-1a was utilized as substrate instead of Z-1a, the reaction still proceeded well to produce the desired product 2a in similar good yield and selectivity (entry 21). This result indicated that the same product could be formed efficiently using an alkene mixture of Z/E isomers as substrate, which is of great significance for practical synthesis due to the quite easy availability of Z/E alkene isomers. Lastly, when the reaction was performed under higher concentration, the yield of product 2a could be further improved to 96% (entry 22).

### Scope of catalytic 1,1-diazidation of alkenes

With the optimized conditions in hand, the scope of alkenes was evaluated (Fig. 2). First, the stilbene derivatives with different electron-rich aryl as migratory group were investigated (Fig. 2, upper part). It was found that the stilbenes with various alkoxyl and aryloxy at the para position of migratory aryl group could be successfully transformed into the corresponding geminal diazides at tetrahedral carbons in good to excellent yields (2a-2f, 68–99%). Interestingly, siloxy-containg substrate 1 g could still undergo the diazidation and rearrangement to afford the desired product 2 g in good yield even in this fluoride-rich system. The stilbenes with methoxy, alkyl, bromo, and chloro groups at the meta position and alkoxyl at the para position were well tolerated under the standard conditions as well (2h-2m, 83–97%). In addition to the substrates bearing alkoxyl group at the para position, the stilbenes with methoxy at the ortho position of migratory aryl group could also produce the desired geminal diazides in good yields despite the influence of steric hindrance (2n-2p, 67–81%). Besides, in comparison with the methoxy-substituted aryl substituents, other electron-rich rings such as fused naphthyl and heteroaromatic indole and thiophene could act as good migratory groups to promote this 1,1-diazidation efficiently (2q-2u, 66–85%). It is noted that the trisubstituted stilbenes as substrates could still give the corresponding geminal diazides in good yields, which clearly broadens the scope of the products (2v-2x, 60–72%). Furthermore, aniline derived alkene could go through the azidation to afford the geminal diazides in 60% yield, but with inseparable by-product vicinal diazides (see Supplementary Fig. 6). When the migratory group was replaced with phenyl or electron-deficient aryl, the vicinal diazides instead of geminal ones turned to be major products or the reaction became messy (see Supplementary Figs. 7–9).

Then, the scope of group R² rather than the migratory aryl group on alkenes was studied (Fig. 2, middle part). To our delight, when

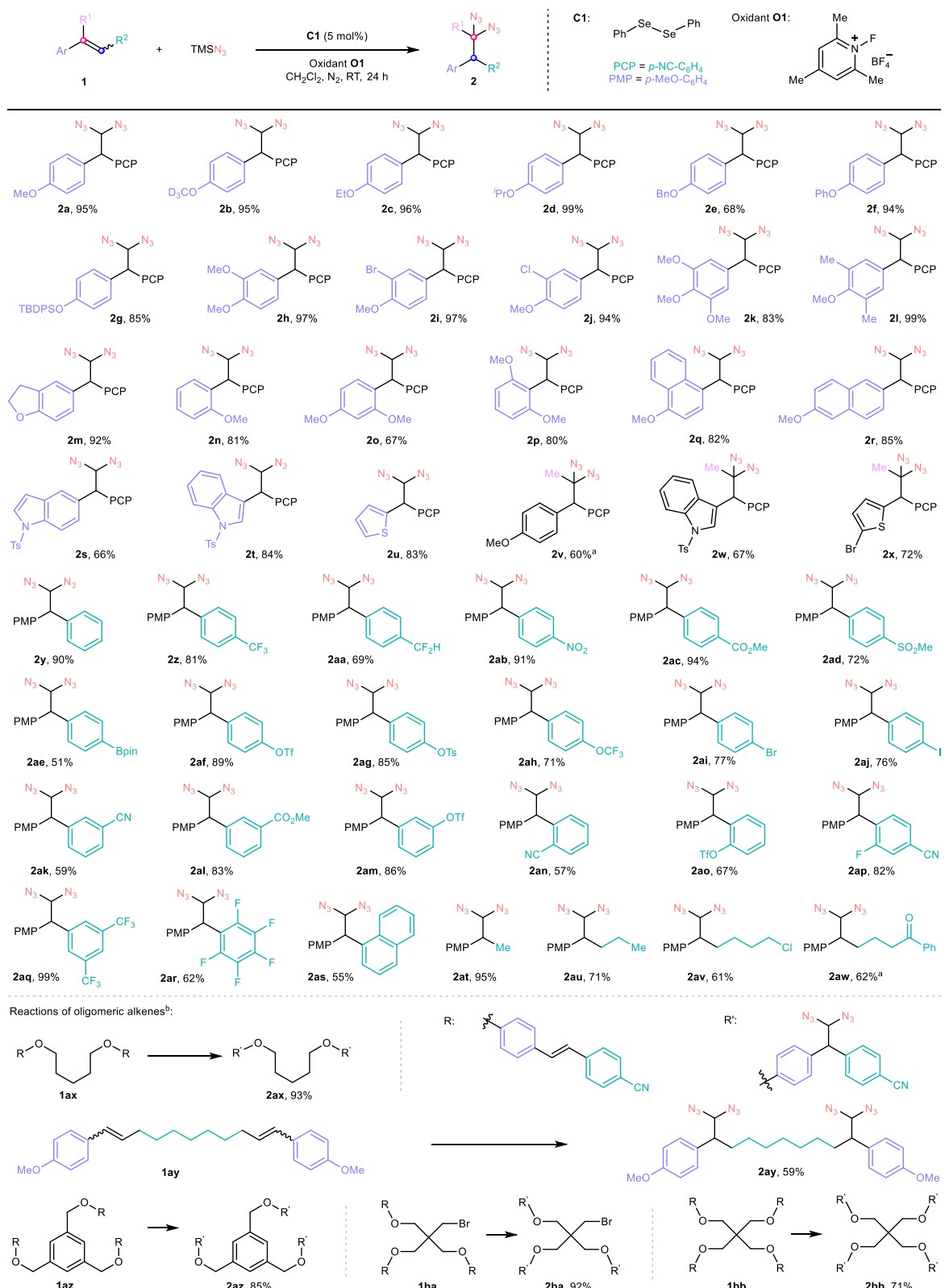

**Fig. 2 | Catalytic 1,1-diazidation of alkenes.** Reaction conditions: **1** (0.10 mmol), azidotrimethylsilane (0.30 mmol, 3.0 equiv), **O1** (0.20 mmol, 2.0 equiv), **C1** (0.005 mmol, 5 mol%), dichloromethane (350 μL), N₂, room temperature, 24 h. ᵃ**C1** (10 mol%). ᵇFor the reactions of oligomeric alkenes, **1ax** (0.05 mmol), **1ay** (0.05 mmol), **1az** (0.0333 mmol), **1ba** (0.0333 mmol), **1bb** (0.025 mmol) were utilized, respectively. PCP *p*-cyanophenyl, PMP *p*-methoxyphenyl, TBDPS *tert*-butyldiphenylsilyl, Ts *p*-tosyl, pin pinacol group, Tf trifluoromethanesulfonyl.

group R² was electron-neutral phenyl or electron-deficient aryl containing an electron-withdrawing group such as trifluoromethyl, difluoromethyl, nitro, ester, sulfonyl, borate, triflate, tosylate, trifluoromethoxy, bromo, and iodo groups at the *para* position, the reactions proceeded smoothly to give the corresponding products in good to excellent yields (**2y**-**2aj**, 51–94%). The groups such as borate, triflate, and bromo and so on are good convertible groups, which provides a good opportunity for the further transformations of products. It is noted that when group R² was *p*-methoxyphenyl group as same as the group on the other side of the double bond, the reaction only gave the corresponding vicinal diazides instead of the desired geminal product. More details about this result will be discussed in the section of mechanistic study. Similarly, other electron-deficient aryl groups bearing *meta* or *ortho* substituents or multi-substituents lead to the formation of the desired products in moderate to excellent yields (**2ak**-**2ar**, 57–99%). To our surprise, when group R² was ring-fused naphthyl, the reaction still gave the desired product (**2as**, 55%). In addition to the stilbene derivatives, β-alkyl-substituted electron-rich styrenes could also tolerate the conditions to generate the desired products in good to excellent yields (**2at**-**2aw**, 61–95%). It is worthy to mention that the desired products could be formed in high yields without any impact even using the alkenes with two Z/E isomers as substrates. This fact offers a convenience for practical synthesis by this method. Interestingly, for the common alkylated substrates **1at**-**1aw**, alkenylated azides generated from possible side reaction via β-H elimination were not observed. This excellent selectivity of migration versus β-H elimination might be ascribed to relatively lower acidity of β-hydrogen on the corresponding selenenylated intermediates. If there was an electron-withdrawing group on the allylic position of the substrates, β-H elimination could occur rather than migratory functionalization (see Supplementary Fig. 6)[54,56]. Besides, the possible side products derived from deoxygenative azidation or oxidative α,α-diazidation of ketone group on alkene **1aw** was not observed, suggesting that this method is a good supplement to traditional deoxygenative azidation and oxidative α,α-diazidation of carbonyl compounds[24–26].

Except the reactions of monoalkene substrates, we wondered whether our method could be suitable for the reactions of oligomeric alkenes to give the corresponding polydiazide products. Pleasedly, the desired bis, tris, and tetra diazides **2ax**-**2bb** could be produced in good to excellent yields when oligomeric alkenes **1ax**-**1bb** were used as substrates (Fig. 2, lower part). In the literature, polydiazides could act as potentially useful building blocks for the rapid construction of linear and poly(triazole) dendrimers via click chemistry[64–66] and potential crosslinking agent compounds for organic light emitting device[67,68]. Obviously, our method provides a good route for the synthesis of polydiazides through the change of carbon framework, which might promote the application of polydiazides in material sciences.

## TGA-DSC analysis and impact sensitivity tests
We studied the TGA-DSC spectrums of representative geminal diazides with relatively lower carbon to nitrogen ratios ($(N_C + N_O)/N_N$) (Fig. 3)[6]. It was found that most products were stable before 118 °C (except **2at**, 84 °C), but gradually decomposed when the temperature was higher. These results revealed that the obtained geminal diazides are thermally stable. Considering relatively faster decomposition rate of **2at**, the impact sensitivity of **2at** was measured by standard fallhammer tests. It was found that the sample was insensitive to impact (IS > 80 J)[69]. Both TGA-DSC analysis and impact sensitivity tests suggested that these products are relatively safe for conventional use.

## Scale-up synthesis and further transformations of the products
To demonstrate the robustness and practicability of the developed method, the scale-up reactions were completed. When the reactions were carried out using alkenes **1a**, **1af**, and **1at** in several hundred milligrams to one gram scale, the corresponding geminal diazides **2a**, **2af**, and **2at** could be still formed in good yields, respectively (Fig. 4, upper part). It is noted that these scale-up reactions could be handled routinely, which suggested that this type of geminal diazides was relatively stable to some degree, even so, other types of diazides might be potentially explosive in the literatures. To describe the synthetic value and disclose the reactivities of geminal diazides, diverse transformations of the products were performed (Fig. 4, lower part). The tolerance of the diazidomethyl moiety in different reactions was first investigated. To our delight, the geminal diazidomethyl group was not affected in palladium-catalyzed Suzuki-Miyaura cross-coupling reactions, hydrolysis of nitriles, and disulfide-catalyzed electrophilic aromatic halogenation. The desired functionalized diazides **4a**-**4c**, **2i**, and **2j** were generated smoothly. These results indicated that this method provides a good opportunity for the synthesis of complicated geminal diazides. Owing to the presence of the geminal diazidomethyl

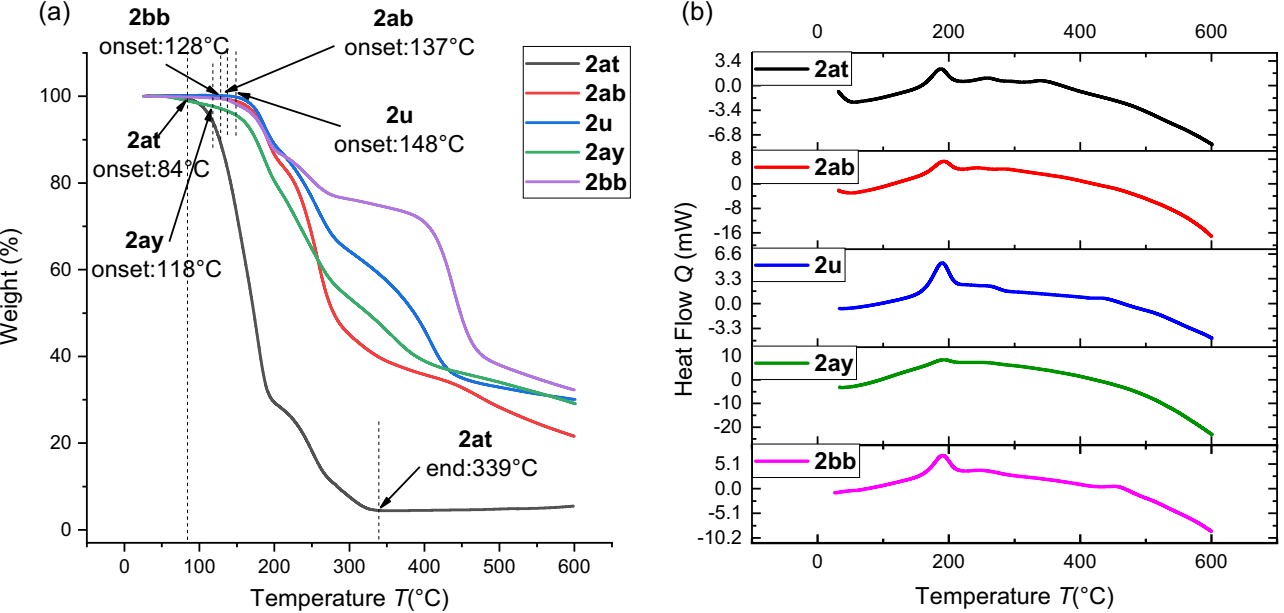

**Fig. 3 | TGA and DSC analysis. a** TGA plots of **2at**, **2ab**, **2 u**, **2ay** and **2bb** (heating rates:10 K·min⁻¹). **b** DSC curves of **2at**, **2ab**, **2 u**, **2ay** and **2bb** (heating rates:10 K·min⁻¹).

**Fig. 4 | Scale-up synthesis and further transformations of the products.** Conditions: **a 2aj**, 4-tolyboronic acid, palladium diacetate, phenylurea, potassium carbonate, methanol+water, RT, 4 h. **b 2aj**, *E*-styrylboronic acid, palladium diacetate, phenylurea, potassium carbonate, methanol+water, RT, 4 h. **c 2a**, hydrogen peroxide, potassium carbonate, dimethyl sulfoxide, 0 °C, 5 min. **d 2a**, *N*-bromosuccinimide, diphenyl disulfide, acetonitrile, RT, 12 h. **e 2a**, *N*-chlorosuccinimide, diphenyl disulfide, acetonitrile, RT, 12 h. **f 2at**, 4-ethynylanisole, copper(II) sulfate pentahydrate, sodium L-ascorbate, *N,N*-dimethylformamide, RT, 12 h. **g 2at**, 1-hexyne, copper(II) sulfate pentahydrate, sodium L-ascorbate, *N,N*-dimethylformamide, RT, 12 h. **h 2a**, dimethyl acetylenedicarboxylate, toluene, 110 °C, 12 h. **i 2a**, 2-

(trimethylsilyl)phenyl triflate, cesium fluoride, acetonitrile, N₂, RT, 12 h. **j 2a**, Erlotinib, cuprous iodide, *N,N*-diisopropylethylamine, acetonitrile, RT, 12 h. **k 2a**, potassium *tert*-butoxide, tetrahydrofuran, N₂, RT, 12 h. **l 2a**, triphenylphosphine, tetrahydrofuran+water, RT, 12 h. **m 2af**, triphenylphosphine, tetrahydrofuran +water, RT, 12 h. **n 2a**, boron trifluoride diethyl etherate, dichloromethane, RT, 12 h. **o 2a**, trimethylsilyl cyanide, boron trifluoride diethyl etherate, dichloromethane, RT, 12 h. **p 2a**, trimethylsilyl isothiocyanate, boron trifluoride diethyl etherate, dichloromethane, RT, 12 h. **q 2a**, triethylsilane, boron trifluoride diethyl etherate, dichloromethane, RT, 12 h. **r 2at**, allyltrimethylsilane, scandium trifluoromethanesulfonate, deuterated dichloromethane, RT, 12 h.

moieties, the products could be converted into various valuable bis-triazoles via click chemistry. Aryl and alkyl alkynes, electron-deficient acetylenedicarboxylate, and benzyne, could undergo double [3 + 2] azide-yne cycloaddition with geminal diazides successfully to produce geminal bistriazoles **4d**-**4f** and bisbenzotriazoles **4 g**. Besides, this click reaction could be applied to the functionalization of alkyne-containing antineoplastic agent Erlotinib to yield bistriazole **4 h**, suggesting that these diazides are good reagents for late-stage modification of drugs. Under the treatment of diazides **2a** with potassium *tert*-butoxide, elimination reaction occurred efficiently to produce alkenyl azide **4i** in good yield. Surprisingly, diarylketones **4j** and **4k** were formed when diazides **2a** and **2af** were treated with triphenylphosphine under Staudinger reduction conditions, respectively. Why were the diarylketones produced in this transformation? The reason might be ascribed to which the formed geminal diamines from the reduction of the azido groups on the geminal diazides undergo hydrolysis to give unstable α,α-diaryl aldehydes, followed by decomposition to form the diarylketones (see Supplementary Discussion)[70,71]. This phenomenon is consistent with the observation that diarylketones were formed when we

tried to prepared the corresponding α,α-diaryl aldehydes by other methods. Interestingly, treatment of **2a** with Lewis acidic boron trifluoride diethyl etherate resulted in the formation of diarylmethyl azide **4 l**. This discovery revealed interesting reactive properties of geminal diazides and is inspiring since it shows that geminal diazidomethyl moiety can act as a formal leaving group in the presence of Lewis acid to facilitate further nucleophilic substitution. Encouraged by this result, we used other nucleophiles for the similar reactions. When **2a** was treated with trimethylsilyl cyanide, trimethylsilyl isocyanate, and triethylsilane in the presence of boron trifluoride diethyl etherate, the corresponding diarylmethyl products **4m**-**4o** were achieved successfully. Similarly, geminal diazides **2at** could be converted into alkene **4p** in the presence of scandium trifluoromethanesulfonate with allyltrimethylsilane. The related mechanism will be discussed in the mechanistic section and Supplementary Information (see Supplementary Discussion). These abovementioned representative transformations clearly showed the great convertibility of geminal diazides. At the same time, these results suggested that geminal diazides can decompose under acidic

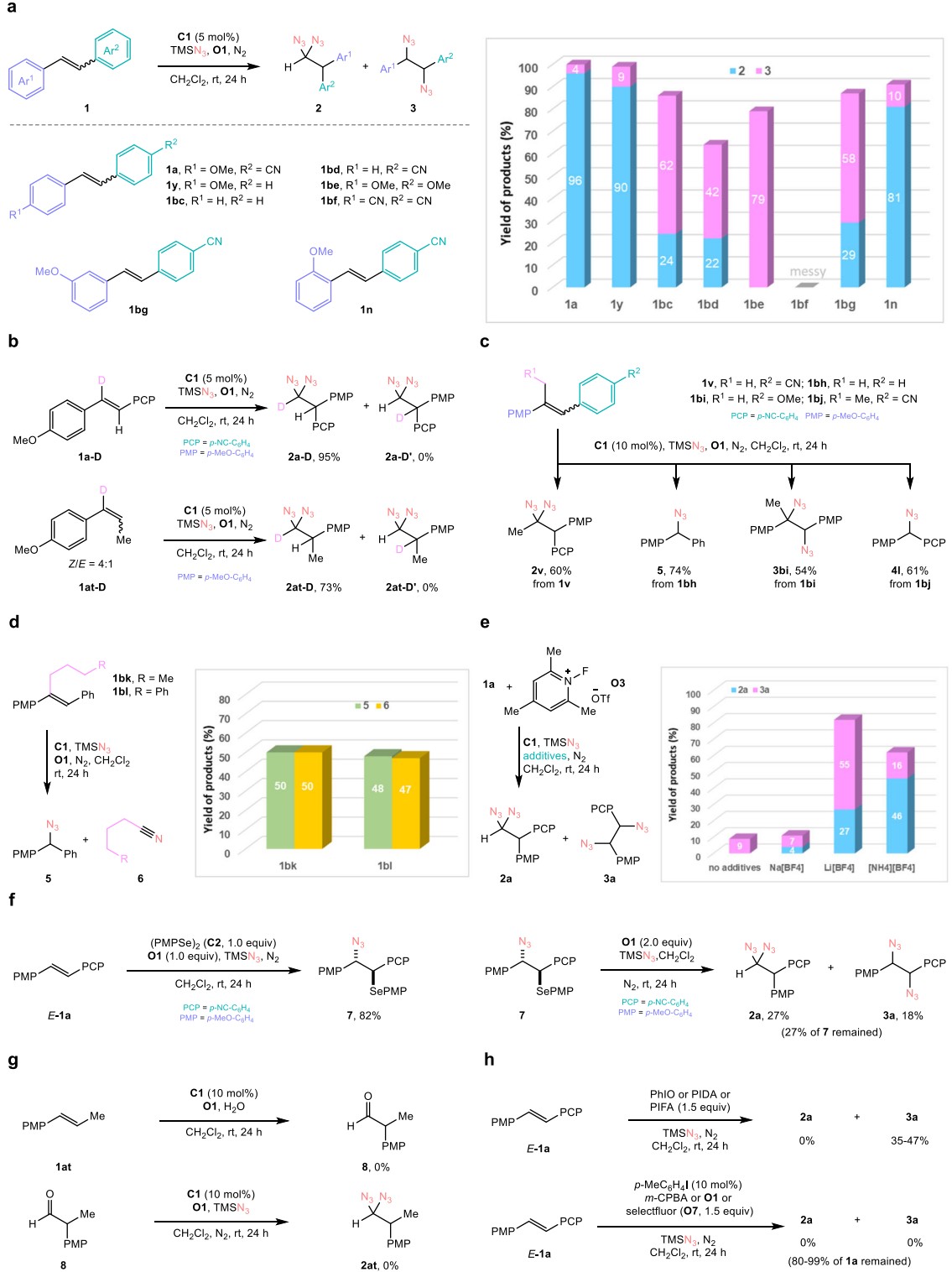

**Fig. 5 | Mechanistic insight. a** Competition between 1,1- vs 1,2-diazidation of stilbene derivatives. **b** Isotope labeling experiments. **c** Diazidation vs carbon deletion reaction of trisubstituted alkenes. **d** Determination of by-products in carbon deletion reaction, **e** Influence of tetrafluoroborates in the reactions. **f** Observation and oxidative azidation of possible intermediates. **g** Studies of exclusion of aldehyde as possible intermediate. **h** In comparison with hypervalent iodine mediated or catalyzed process. PMP *p*-methoxyphenyl, PCP *p*-cyanophenyl.

conditions. However, acid-promoted synthesis of diazides from the corresponding aldehydes is a known tranditional strategy. This means that the target molecules in this work are not easily accessible by traditional methods, and this developed method is a good complement.

## Mechanistic insights
To elucidate the mechanism of this diazidation, control experiments and related reactions have been conducted. Competition between 1,1- and 1,2-diazidation of different 1,2-disubstituted stilbene derivatives was first studied (Fig. 5a). It was found that two competitive reactions

largely depended on the electron richness of the migratory group. When the migratory aryl group was electron-rich enough and more electron-rich than the group on the other side of the double bond, the 1,1-difunctionalization occupied a dominant position. Otherwise, the vicinal diazides would be the major products (**1a, 1y, 1n** vs **1bc, 1bd, 1bg**). This result suggested that the reaction might go through a carbocation intermediate. However, if two electron-rich aryl groups connecting to the double bond were electron-rich enough, only vicinal diazide was observed as products (**1be → 3be**, 79%), suggesting that the intermediate prior to aryl migration is more stable than the one after aryl migration during the reaction of this type of substrates. These results revealed that the stilbenes with one electron-rich aryl group and one electron-neutral or electron-deficient aryl group prefer to form geminal diazides. To further confirm that the reaction went through an electron-rich aryl group migration, the deuterium-labeled stilbene **1a**-D and styrene **1at**-D were utilized as substrates under standard reaction conditions (Fig. 5b). Products **2a**-D and **2at**-D were exclusively formed via *p*-methoxyphenyl migration, but products **2a**-D′ and **2at**-D′ were not observed via *p*-cyanophenyl or methyl migration.

Then, competitive side reactions of trisubstituted stilbene derivatives were studied (Fig. 5c). It was found that different trisubstituted alkenes gave different products such as vicinal and geminal diazides and others. Trisubstituted stilbene **1v** afforded geminal diazides **2v**. However, when the *p*-cyanophenyl group on **1v** was replaced by the phenyl group or the methyl group on **1v** was replaced by the ethyl group, diarylmethyl azides **5** and **4l** were formed probably through aryl migration and carbon deletion. When the *p*-cyanophenyl group on **1v** was replaced by the *p*-methoxyphenyl group, vicinal diazide **3bi** was formed in the reaction similar to the one of electron-rich stilbene **1be**. These results indicated that the oxidative azidation of trisubstituted alkenes is sensitive to the structure of the substrates. Moreover, the byproducts in the azidative carbon delection reaction were determined as nitriles by using trisubstituted stilbenes **1bk** and **1bl** as substrates, and the amounts of nitriles were nearly equal to the amount of the corresponding diarylmethyl azides **5** (Fig. 5d). This result provides evidence for the reaction mechanism.

To prove the assumption that the tetrafluoborate anion could promote the reaction, **O3** containing a triflate as anion was utilized as oxidant in the reaction of *E*−**1a** in combination with other inorganic tetrafluoborate salts as additives. As depicted in Fig. 5e, the total yield of geminal diazides **2a** and vicinal diazides **3a** could be improved largely when lithium tetrafluoborate (LiBF$_4$) or ammonium tetrafluoborate (NH$_4$BF$_4$) was utilized as additive. This result indicated that the tetrafluoborate anion indeed improved the reaction, possibly because of the reason mentioned in the section of reaction condition optimization.

To understand more about the reaction mechanism, stilbene *E*−**1a** was treated with 1.0 equivalent of di(*p*-methoxyphenyl) diselenide **C2** in the presence of equal equivalent of oxidant **O1** and excess azidotrimethylsilane (Fig. 5f, left). Selenenylated intermediate **7** was isolated in 82% yield. Intermediate **7** could be converted into geminal diazide **2a** and vicinal diazide **3a** in the presence of **O1** and azidotrimethylsilane (Fig. 5f, right), confirming that **7** is a possible intermediate. It is noted that the amount of vicinal diazides increased in this stoichiometric deselenenylation process, which might be ascribed to the concentration effect since the ratio of oxidants and azidotrimethylsilane to the intermediate under catalytic conditions is much higher than the one under stoichiometric conditions. In consideration of that deoxygenative azidation of aldehydes could yield geminal diazides[28], we wondered whether aldehyde was a possible intermediate. When the reaction was carried out using water instead of azidotrimethylsilane, alkene *E*-**1at** could not afford aryl-migrated aldehyde **8** (Fig. 5g, top). In addition, when aldehyde **8** was utilized as substrate under the standard oxidative conditions, no desired

geminal diazide **2at** was formed (Fig. 5g, bottom). These results suggested that aldehyde is not the intermediate in this reaction.

At last, the 1,1-diazidation of alkenes was studied by the system of hypervalent iodine in consideration of whether hypervalent iodine could mediate or catalyze aryl migratory functionalization of alkenes (Fig. 5h)[35,36]. It was found that hypervalent iodine system could not furnish the desired geminal diazides no matter in the fashion of stoichiometry or catalysis. This result reflected that the advantages and uniqueness of redox-active selenium catalysis.

On the basis of the mechanistic studies and the previous report[50], the plausible mechanisms are proposed in Fig. 6. First, catalyst precursor diselenide **C1** is oxidized by **O1** to form the real catalyst phenylselenium tetrafluoroborate together with phenylselenium fluoride and 2,4,6-collidine. At the same time, a small part of oxidant **O1** decomposes to the fluorinated pyridine derivative **9**[72], of which polarity is similar to some of geminal diazide products. Compound **9** can be easily removed by washing with diluted hydrochloric acid after the reaction. Subsequently, phenylselenium tetrafluoroborate interacts with **1a** to form the seleniranium intermediate **I**. The tetrafluoborate anion acts as the activator of azidotrimethylsilane via fluoro-silicon interaction to accelerate the release of azide anion. The anion serves as a nucleophile to attack the carbon connecting to the *p*-methoxyphenyl group on seleniranium **I**, resulting in yielding the selenenylated intermediate **II**, trimethylfluorosilane, and trifluoroborane. It is noted that the attack of the azide group toward intermediate **I** is in excellent regioselectivity because the reactive site attached by the *p*-methoxyphenyl group is more electrophilic than the site attached by the *p*-cyanophenyl group[52]. After oxidation of **II** by **O1**, selenium (IV) species **III** is generated[73,74]. The species **III** can give phenylonium ion intermediate **IV** after the attack from intramolecular vicinal electron-rich aryl group and release phenylselenium fluoride. phenylselenium fluoride might react with trifluoroborane to regenerate phenylselenium tetrafluoroborate. An equilibrium might exist between **IV** and azide-stabilized intermediate, azidocarbenium ion **V**[75]. After a second azidation with another molecular azidotrimethylsilane, the desired geminal diazide **2a** is formed. During the reaction, intermediate **III** might also go through a minor path to afford intermediate **VI** with a structure of benzyl carbocation, followed by the attack of **VI** by azidotrimethylsilane to form vicinal diazide **3a**. A path between intermediate **VI** and **IV** is also possible. For this diazidative reaction of alkenes containing at least one electron-rich aryl group, the 1,1-/1,2- selectivity depends on the relative stability of intermediate **V** and **VI**. For the substrates in Fig. 2, the azide-stabilized intermediate **V** is more stable than electron-deficient aryl-stabilized intermediate **VI**, resulting in formation of geminal diazides as major products. For the reaction of bi(*p*-methoxyphenyl) stilbene derivatives **1be** and **1bi** in Fig. 5a, c, the formed intermediate **VI** is more stable because the carbocation is connected to the electron-rich *p*-methoxyphenyl group, resulting in yielding vicinal diazides as major products. Although we cannot rule out that our reaction goes through a selenium(IV) species initiated process[76], we are inclined to believe that a selenium(II) species initiates the reaction at the beginning since temperature-sensitive selenium(IV) polyazides might be difficult to exist under the reaction temperature[77]. Another interesting reaction is the carbon delection of some trisubstituted stilbenes. As depicted in Fig. 6b using stilbene **1bk** as an example, phenylonium ion intermediate **VII** might be formed in a similar way to the formation of **IV**. Probably due to the steric hindrance of the multisubstituted cyclopropane structure, intermediate **VII** might decompose to *p*-quinone methide intermediate **VIII** via release of one molecule of nitrogen and nitrile (path A)[78,79]. An equilibrium might exist between **VIII** and diarylmethyl carbocation **IX**, which could be attacked by azide anion to afford diarylmethyl azide **5**[80]. **VII** might also

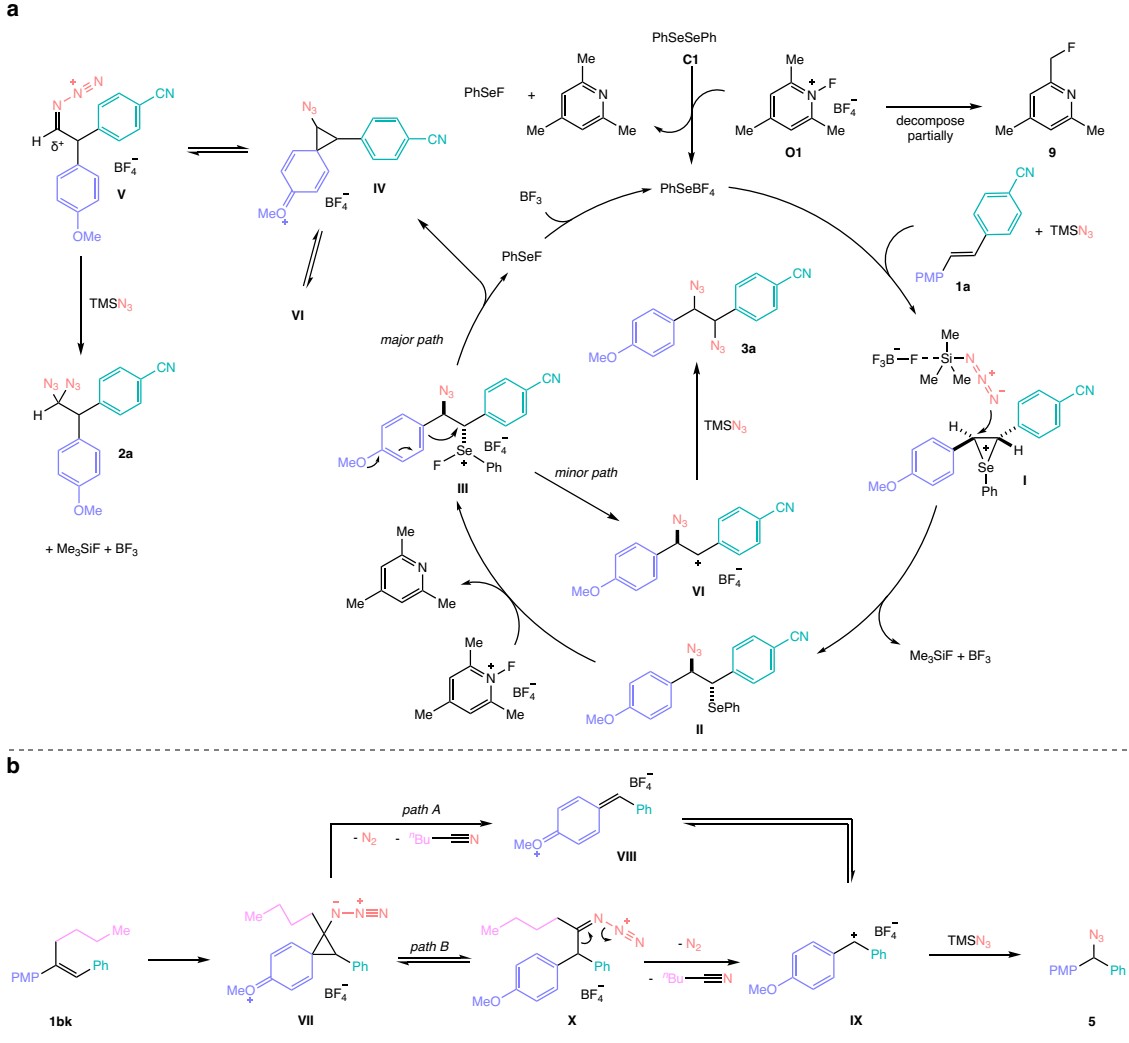

**Fig. 6 | Proposed mechanism. a** Formation of **2a** via 1,1-diazidation and **3a** via 1,2-diazidation. **b** Formation of **5** via carbon deletion. PMP *p*-methoxyphenyl.

isomerize to intermediate **X** first, followed by release of nitrogen and nitrile to yield **IX** (path B). It is worth mentioning that similar products can be formed by treatment of geminal diazides **2a** and **2at** with Lewis acid boron trifluoride diethyl etherate and scandium trifluoromethanesulfonate. It is possible that the similar intermediate **VII** could be formed via the leaving of the azido group promoted by Lewis acid and the attack of the neighboring *p*-methoxyphenyl group. After the release of nitrogen and nitrile and the attack by exogenous nucleophiles, products **4l-4p** are generated, as shown in Fig. 4.

In conclusion, we have developed an efficient approach for the synthesis of geminal diazides via redox-active selenium catalyzed 1,1-diazidation and rearrangement of alkenes. A series of geminal diazides are formed through a carbon framework change in a broad substrate scope. The formed diazides are relatively safe by TGA-DSC analysis and impact sensitivity tests. They can serve as good precursors for preparation of various valuable molecules. Mechanistic studies revealed that a selenenylation-deselenenylation followed by an 1,2-aryl migration process is involved in the reactions, and the 1,1-difunctionalization benefits from not only the electron-rich migratory aryl group but also the stability of the azidocarbenium intermediate after migration. This work provides a strategy for the synthesis of diazides, and represents an interesting case of 1,1-difunctionalization of alkenes by selenium catalysis, which is greatly complementary to the fields of alkene chemistry and group 16

element catalysis. Development of new reactions of alkenes by group 16 element catalysis is ongoing in our laboratory.

## Methods

### General procedure for the catalytic 1,1-diazidation of alkenes
To a dry 4 mL vial equipped with a stir bar were added alkene **1** (0.1 mmol, 1.0 equiv), **O1** (45.4 mg 0.2 mmol, 2.0 equiv), and **C1** (1.6 mg, 0.005 mmol, 5 mol%) successively. Then the vial was transferred into glovebox. After that dry dichloromethane (350 μL) and azido-trimethylsilane (39.5 μL, 0.3 mmol, 3.0 equiv) were added successively under inert gas, the vial was capped and removed from glovebox. The reaction was performed at room temperature for 24 h. Then the resulting mixture was quenched with diluted hydrochloric acid (1.0 mL, 1 M) and extracted with dichloromethane (3 mL × 3). The combined organic layers were washed with brine, dried over sodium sulfate, and concentrated under reduced pressure. The residue was purified by preparative thin layer chromatography to give the desired geminal diazides **2**.

## Data availability
The data supporting the finding of this study are available in this article and the Supplementary Information. Crystallographic data for structures **2e** has been deposited at the Cambridge Crystallographic Data Centre, under deposition number CCDC2253572. Copies of the data can be obtained free of charge via https://www.ccdc.cam.ac.uk/

structures/. All other data are available from the corresponding author upon request.

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

## Acknowledgements

Financial support was provided by the National Natural Science Foundation of China (Grant Nos. 22371308, 22171292 and 22101304) and Guangdong Basic Research Center of Excellence for Functional Molecular Engineering. We are grateful to Dr. Xue-Wen Zhang for the single-crystal X-ray structure analysis of **2e**.

## Author contributions

X.Z. and L.L. conceived and directed the project. L.L. and X.X. prepared most of alkene substrates. W.Q. performed condition evaluation and most experiments about the exploration of substrate scope. W.Q. and L.L. performed safety evaluation, most synthetic applications, mechanistic studies, data analysis, and the preparation of Supplementary Information. H.H. performed the additional experiments with respect to substrate scope and synthetic applications. Y.X. performed initial exploration of this project. L.L. prepared the manuscript, and X.Z. revised the manuscript and Supplementary Information. All authors discussed the results and commented on the paper.

## Competing interests

The authors declare no competing interests.

## Additional information

Lihao Liao or Xiaodan Zhao.

