## [Peer Review File · Nature Communications]

Catalytic 1,1-diazidation of alkenesREVIEWER COMMENTS

Reviewer #1 (Remarks to the Author):

The manuscript by Zhao, Liao, and co-workers describes a new and impressive catalytic approach to 1,1-diazides via an aryl group migration by the combination of an azide source (TMSN₃), diselenide catalyst (Ph₂Se₂) and oxidant (fluoropyridinium salt). The desired products are achieved in high yields. The scope of the reaction is relatively broad in a wide range of electron-rich aryl alkene substrates. Besides the successful substrates, a collection of ineffective substrates is presented in the supporting information, providing a nice summary of the limitations. The authors evaluated the safety of products by TGA-DSC analysis and impact sensitivity tests. The applicability of the obtained products is demonstrated by different transformations. The mechanistic studies provide important insights into the overall process of the reaction. As mentioned in the manuscript by the authors, the known methods usually focus on the synthesis of 1,2-diazides in azide field. In this work, the authors developed a method of catalytic migratory 1,1-diazidation of alkenes, which is greatly complementary to the fields of azide and alkenes. Besides, this work is also a good example of selenium catalysis, different with the known selenium catalysis. Overall, this work is extraordinary, well documented in details with high quality of the SI.

Considering the high novelty and the significance of this work, I strongly recommend to accept this manuscript for publication in this journal. Some minor questions and suggestions need to be addressed:

(1) Using disulfide and ditelluride as the catalyst precursors, the reaction did not work at all. The authors should mention the reason in the text in order to let the readership to learn the difference between them.

(2) In the description of synthetic application, the “epresentative” in the sentence “These above-mentioned epresentative transformations clearly showed the great convertibility...” should be revised as “representative”.

(3) In the description of Fig.3b, the prefix “p-” in “p-methoxyphenyl migration” and “p-cyanophenyl” should be italic.

(4) In the description of Fig.3g, it's more reasonable to cite reference “28” instead of “17,18” in the sentence of “In consideration of that deoxygenative azidation of aldehydes

could yield geminal diazides^{17,18}...” .

(5) In the description of Fig.3h, it's more reasonable to cite references “35,36” instead of “39,40” in the sentence of “...whether hypervalent iodine could mediate or catalyze aryl migratory functionalization of alkenes (Fig. 3h)^{39,40}...” .

(6) The name of last author in ref. 3, “G, Yin”, should be revised as “Yin, G.”.

(7) The ref. 42 has exact volume and page numbers now.

Reviewer #2 (Remarks to the Author):

Review of “Catalytic 1,1-diazidation of alkenes”

The authors report a Se-catalyzed 1,1-diazidation of alkenes. To get both azides on the same carbon atom, an electron-rich aryl substituent on the original alkene must undergo 1,2-migration. Based on the reported substrate scope, this rearrangement only takes place with aryl groups that are substituted with an alkoxy group, so that an intermediate carbocation can undergo a phenonium ion rearrangement, otherwise 1,2-diazidation takes place instead. What this means is that only stilbenes (and a small number of styrenes) in which one aryl group has an alkoxy group can be employed as substrates, thereby generating 1,1-diazido-2,2-diarylethanes. The reaction is catalyzed by diphenyl diselenide and uses an N-fluoropyridinium as the oxidant in conjunction with TMSN₃ as the azide source. The yields are high, but given the limitations on substrate described above, the authors report a far too large number of substrates (over 50, but only varied in the nature of the non-electron-rich aryl group). The authors report a number of functionalizations of the products, though most are routine (cross-couplings, Huisgen cyclizations), or of limited utility (formation of diarylmethyl derivatives). They also report a number of mechanistic experiments that conclusively establish the key mechanistic details, notably the phenonium rearrangement and the role of BF₄ in promoting the reaction.

Overall, this is an extremely thorough exploration of this selenium-catalyzed reaction. They have left no stone unturned in terms of substrate scope or mechanistic investigation, and this would serve as a comprehensive manuscript for this chemistry.

On the other side, the transformation is similar to others reported from this lab and others, with the extra twist of the phenonium rearrangement. Indeed, the catalyst and oxidant are identical to those used many times by this lab, and there isn't anything fundamentally new in the mechanism they establish, and it doesn't immediately provide new insight into selenium catalysis in general, other than the idea that BF₄ can help promote reactivity of trimethylsilyl pronucleophiles. The limitations imposed on the substrate scope by their mechanism also significantly limit the impact of this reaction. The inclusion of 50 examples doesn't change that. Finally, it's also unclear what utility this reaction or its products might have. Since the 1,1-diazido functional group is rare, and azides themselves are rarely the desired final products, their subsequent reactivity is important. The most common reaction, reduction to the amine, would provide products more easily obtained in other ways, and other transformations also provide products that are easier to obtain in other ways. Consequently, almost none of the functionalization reactions they report would appear to be immediately useful. Perhaps the only described application with potential utility is to use Huisgen cyclization reactions to prepare polymers or oligomers. But even here it is not clear what advantage the use of geminal diazides provides over other kinds of polyazides that are potentially easier to prepare.

In reading the authors' response to the initial decision, it seems to me that both the editor and the authors are aware of both the positive and negative aspects of this manuscript I explained above. I don't find the authors' response to be especially compelling in rebutting the negative issues pointed out above. Not to minimize the difficulty of developing this reaction or the thoroughness of their work, but I still find this to be a relatively niche application of selenium catalysis whose impact on synthetic chemistry is likely to be minor at this point.

Reviewer #3 (Remarks to the Author):

Manuscript ID: NCOMMS-23-42971A-Z

Title: Catalytic 1,1-diazidation of alkenes

Author(s): Wangzhen Qiu, Lihao Liao, Xinghua Xu, Hongtai Huang, Yang Xu, and Xiaodan

Zhao

General Comments:

Zhao and coworkers developed an interesting method that provides a comprehensive exploration of catalytic 1,1-diazidation of alkenes, shedding light on a barely explored but valuable aspect of alkene functionalization. Significant contributions to the field of organic chemistry, specifically in the development of producing geminal diazides. Because this is an important area of research due to the potential applications of azides in synthetic chemistry. The authors have demonstrated the catalytic 1,1-diazidation of alkenes with well-defined conditions and have provided detailed information about the scale-up synthesis. Moreover, this study's thorough investigation of substrate scope, synthetic applications, and mechanistic insights significantly advances the field. The inclusion of crystallographic data for structures 2e adds credibility to the findings. Overall, the work presents novel findings in the catalytic 1,1-diazidation of alkenes and the synthetic applications of geminal diazides, making it a valuable contribution to the field of organic chemistry. Furthermore, the manuscript has been well-prepared and clearly presented, showcasing the expertise of the authors in this field. The supporting information is also well prepared and easy to follow. However, it would be beneficial for the authors to provide additional details regarding the substrate scope, reaction mechanism. Addressing these points would enhance the manuscript's scientific impact and provide a more comprehensive understanding of the reported research.

Recommendation: Publication in Nature Communications is supported after Major revision.

Specific comments:

- 1) As reported by Wolloch et al. (ANNALEN DER CHEMIE-JUSTUS LIEBIG, 1975, 2339). 1,3-dienes can undergo group migration to yield diazides albeit with low yield. This reviewer wonders what are the biggest difference and challenge when compared to Wolloch's work and this manuscript.
- 2) For alkylated substrates, such as 1at-aw, the beta-H elimination often happened to form alkenylated azides. Are there any such side products observed for reactions with 1at-aw? If not, the excellency in selectivity should be discussed?

- 3) The substrate scope is somehow limited to arylated alkenes. Can 1,3-dienes be used as the substrates or other substrates tolerated?
- 4) PhSeSePh is a common catalyst for many reactions. In some of them, the Se(IV) species was supposed to be the reactive catalysts, for example, in Denmark's paper, *Nature Chemistry*, 2015, 7, 146. In this paper, PhSeBF₄ was regarded as the real catalyst. Could the authors provide more explanation on why PhSeBF₄ was the active catalyst?
- 5) In Fig.4 nucleophilic attack of species (I) by TMSN₃ to form species (II) was proposed. However, this reviewer doubt that the carbon attached with PCP should be more electrophilic. Is this correct?
- 6) Migratory 1,1-difunctionalization of alkenes is an interesting strategy for the synthesis of geminal functionalized compounds. Instead of discussion 1,1-difluorination, key literatures for 1,1-diborylalkanes from terminal alkenes should also be mentioned and cited.

Thank all the reviewers very much for their comments to help us to improve the quality of our manuscript. According to the suggestions and questions, we have made some changes in our manuscript and supplementary information. In the following is our point-by-point response to the comments.

1. Response to Reviewer 1's comments:

The reviewer pointed out “*The manuscript by Zhao, Liao, and co-workers describes a new and impressive catalytic approach to 1,1-diazides via an aryl group migration by the combination of an azide source (TMSN₃), diselenide catalyst (Ph₂Se₂) and oxidant (fluoropyridinium salt). The desired products are achieved in high yields. The scope of the reaction is relatively broad in a wide range of electron-rich aryl alkene substrates. Besides the successful substrates, a collection of ineffective substrates is presented in the supporting information, providing a nice summary of the limitations. The authors evaluated the safety of products by TGA-DSC analysis and impact sensitivity tests. The applicability of the obtained products is demonstrated by different transformations. The mechanistic studies provide important insights into the overall process of the reaction. As mentioned in the manuscript by the authors, the known methods usually focus on the synthesis of 1,2-diazides in azide field. In this work, the authors developed a method of catalytic migratory 1,1-diazidation of alkenes, which is greatly complementary to the fields of azide and alkenes. Besides, this work is also a good example of selenium catalysis, different with the known selenium catalysis. Overall, this work is extraordinary, well documented in details with high quality of the SI. Considering the high novelty and the significance of this work, I strongly recommend to accept this manuscript for publication in this journal. Some minor questions and suggestions need to be addressed*” Thanks for this positive comment. The reviewer raised some questions and suggestions as follows:

(1) The reviewer pointed out “(1) Using disulfide and ditelluride as the catalyst precursors, the reaction did not work at all. The authors should mention the reason in the text in order to let the readership to learn the difference between them.”

Thank the reviewer very much for pointing out. We added the following description to the discussion of different dichalcogenide catalyst precursors in the manuscript: “*It was found that the reactions did not work at all when disulfide and ditelluride were utilized as catalyst precursors. For the former, it might be ascribed to the difficulty for the generation of electrophilic sulfur catalyst since thiolated compounds could be oxidized by N-F reagents to produce relatively stable sulfoxides or sulfones⁶⁰. For the latter, the generated high valent tellurium species was relatively robust and might be unreactive toward alkenes⁶¹.*” New reference 60 (Xu, X. et al. Selective synthesis of sulfoxides and sulfones via controllable

oxidation of sulfides with *N*-fluorobenzenesulfonimide, *Org. Biomol. Chem.* **19**, 8691–8695 (2021)) and reference 61 (Bornemann, D. et al. Pentafluoro(aryl)- λ^6 -tellanes and tetrafluoro(aryl)(trifluoromethyl)- λ^6 -tellanes: From SF₅ to the TeF₅ and TeF₄CF₃ groups, *Angew. Chem. Int. Ed.* **58**, 12604–12608 (2019)) were added, and the order of subsequent references changed accordingly.

(2) The reviewer pointed out “(2) *In the description of synthetic application, the “epresentative” in the sentence “These above-mentioned epresentative transformations clearly showed the great convertibility...” should be revised as “representative”.*”

We made change in the manuscript.

(3) The reviewer pointed out “(3) *In the description of Fig.3b, the prefix “p-” in “p-methoxyphenyl migration” and “p-cyanophenyl” should be italic.*”

We made changes in the manuscript.

(4) The reviewer pointed out “(4) *In the description of Fig.3g, it’s more reasonable to cite reference “28” instead of “17,18” in the sentence of “In consideration of that deoxygenative azidation of aldehydes could yield geminal diazides17,18...” .*”

We cited the reference 28 instead of references 17 and 18 in the sentence in the manuscript.

(5) The reviewer pointed out “(5) *In the description of Fig.3h, it’s more reasonable to cite references “35,36” instead of “39,40” in the sentence of “...whether hypervalent iodine could mediate or catalyze aryl migratory functionalization of alkenes (Fig. 3h)39,40...” .*”

Taking the suggestion, we made change in the manuscript.

(6) The reviewer pointed out “(6) *The name of last author in ref. 3, “G, Yin”, should be revised as “Yin, G.”.*”

Taking the suggestion, we made revision in the manuscript.

(7) The reviewer pointed out “(7) *The ref. 42 has exact volume and page numbers now.*”

We added the exact volume and page numbers in the ref. 42 in the manuscript.

2. Response to Reviewer 2’s comments:

The reviewer pointed out “*The authors report a Se-catalyzed 1,1-diazidation of alkenes. To get both azides on the same carbon atom, an electron-rich aryl substituent on the original alkene must undergo 1,2-migration. Based on the reported substrate scope, this rearrangement only takes place with aryl groups that are substituted with an alkoxy group, so that an intermediate carbocation can undergo a phenonium ion rearrangement, otherwise 1,2-diazidation takes place instead. What this means is that only stilbenes (and a small number of styrenes) in which one aryl group has an alkoxy group can be employed as substrates, thereby generating 1,1-diazido-2,2-diarylethanes. The reaction is catalyzed by diphenyl diselenide and uses an N-fluoropyridinium as the oxidant in conjunction with TMSN₃ as the azide source. The yields are high, but given the limitations on substrate described above, the authors report a far too large number of substrates (over 50, but only varied in the nature of the non-electron-rich aryl group). The authors report a number of functionalizations of the products, though most are routine (cross-couplings, Huisgen cyclizations), or of limited utility (formation of diarylmethyl derivatives). They also report a number of mechanistic experiments that conclusively establish the key mechanistic details, notably the phenonium rearrangement and the role of BF₄ in promoting the reaction.*

Overall, this is an extremely thorough exploration of this selenium-catalyzed reaction. They have left no stone unturned in terms of substrate scope or mechanistic investigation, and this would serve as a comprehensive manuscript for this chemistry.

On the other side, the transformation is similar to others reported from this lab and others, with the extra twist of the phenonium rearrangement. Indeed, the catalyst and oxidant are identical to those used many times by this lab, and there isn't anything fundamentally new in the mechanism they establish, and it doesn't immediately provide new insight into selenium catalysis in general, other than the idea that BF₄ can help promote reactivity of trimethylsilyl pronucleophiles. The limitations imposed on the substrate scope by their mechanism also significantly limit the impact of this reaction. The inclusion of 50 examples doesn't change that. Finally, it's also unclear what utility this reaction or its products might have. Since the 1,1-diazido functional group is rare, and azides themselves are rarely the desired final products, their subsequent reactivity is important. The most common reaction, reduction to the amine, would provide products more easily obtained in other ways, and other transformations also provide products that are easier to obtain in other ways. Consequently, almost none of the functionalization reactions they report would appear to be immediately useful. Perhaps the only described application with potential utility is to use Huisgen cyclization reactions to prepare polymers or oligomers. But even here it is not clear what advantage the use of geminal diazides provides over other kinds of polyazides that are potentially easier to prepare.

In reading the authors' response to the initial decision, it seems to me that both the editor and the authors are aware of both the positive and negative aspects of this manuscript I explained above. I don't find the authors' response to be especially compelling in rebutting the negative issues pointed out above. Not to minimize the difficulty of developing this reaction or the thoroughness of their work, but I still find this to be a relatively niche application of selenium catalysis whose impact on synthetic chemistry is likely to be minor at this point." Thanks for the comments.

Based on the above comments, there are several points we would like to give more explanations:

(i) The reviewer pointed out that only stilbenes (and a small number of styrenes) on which one aryl group has an alkoxy group can be employed as substrates, which is a limitation for this work.

Thank the reviewer for this concern. Actually, in addition to alkenes bearing alkoxy group substituted aryl group, substrates with other migratory groups, such as indolyl and thienyl heteroaryl groups, could also produce the desired 1,1-diazides smoothly (see Table 2, **2s-2u**, **2w-2x**). Furthermore, substrates **1bc** and **1bd** with neutral phenyl as migratory group could also yield the migratory products, albeit in relative lower yields (Fig 3a). For these substrates, owing to the similar polarities between 1,1-diazides and 1,2-diazides, the separation of these two types of products is difficult. Furthermore, substrates with aniline and benzofurans as migratory groups could also generate the desired 1,1-diazides in good yields, albeit with similar difficulties in separation of the byproducts (see section 4.4 in the Supplementary Information). Generally, the presence of electron-rich aryl groups as migratory groups not only makes our reaction more efficient, but also makes the purification of products easier.

In the literature, success of some elegant works benefits from the use of substrates containing electron-rich aryl groups, especially alkoxy aryl substituents. Please see the relevant references: *Science*, **361**, 1369–1373 (2018); *Angew. Chem. Int. Ed.* **48**, 7094–7097 (2009); *Angew. Chem. Int. Ed.* **52**, 7151–7155 (2013); *Angew. Chem. Int. Ed.* **55**, 4748–4752 (2016); *Angew. Chem. Int. Ed.* **60**, 1861–1868 (2021). In comparison with the works in the literature, the scope of electron-rich aryl groups in our work is broader and more diverse generally.

(ii) The reviewer pointed out “there isn't anything fundamentally new in the mechanism they establish, and it doesn't immediately provide new insight into selenium catalysis in general, other than the idea that BF₄ can help promote reactivity of trimethylsilyl pronucleophiles”.

Thank the reviewer for this comment. Since Sharpless disclosed selenium-catalyzed chlorination of olefins via a selenylation-deselenylation process in 1979 for the first time (*J. Org. Chem.* **44**, 4204–4208 (1979)), this electrophilic selenium catalysis was gradually applied to

allylic functionalization, alkenyl sp^2 C-H functionalization, and 1,2-difunctionalization of alkenes by utilizing different alkene substrates, additives, nucleophiles, and oxidants. As claimed in this manuscript, this selenium catalysis has not been applied to 1,1-difunctionalization of alkenes as well as the functionalization of alkenes with azides as nucleophiles prior to this work. However, the boundary of this selenium catalysis is unknown, and needs to be explored. To explore the boundary, the impact factors on this catalysis need to be understood. Actually, the success of the functionalization of alkenes by this catalysis depends on the alkene substrates, additives, nucleophiles, and oxidants. Possibly, they have an impact on the formation of important seleniranium ion intermediate and the deselenenylation process to affect the whole transformation. Once the seleniranium ion intermediate is formed, suitable nucleophile is needed to attack the seleniranium moiety. For this transformation we developed, the seleniranium ion intermediate can be formed smoothly in the presence of $TMSN_3$. Then the azide group attacks the seleniranium moiety to give the selenenylated intermediate. Subsequently, during the following step of deselenenylation, there exist three possible pathways: β -hydrogen elimination, the substitution by the azide group, and the substitution by migratory aryl group. In this transformation, the substitution by migratory aryl group plays a dominant role. We consider that these details are important and new aspects in electrophilic selenium catalysis. They will benefit the design of new reactions by selenium catalysis. It is noted that the tetrafluoroborate is important to activate the trimethylsilyl pronucleophiles for the transformation. Otherwise, the reaction would not take place efficiently. These mentioned points are inspiring and inputs to the mechanism of selenium catalysis although they do not change the selenylation-deselenenylation process. They cause the reaction to occur, which showcases that this work is a great progress since it not only provides a new transformation (“1,1-difunctionalization”), but also introduces new nucleophile (“azides”) into selenium redox catalysis.

(iii) The reviewer pointed out that most functionalizations of the products are routine (cross-couplings, Huisgen cyclizations), or of limited utility (formation of diarylmethyl derivatives), and also commented that it was unclear what utility this reaction or its products might have and almost none of the functionalization reactions would appear to be immediately useful.

Thank the reviewer for the comment. We agree with the reviewer to some extent. But some important points and potential value for this work cannot be ignored. In general, 1,1-diazides are a neglected class of compounds in literature. Owing to the lack of efficient synthetic methods for the synthesis of diverse 1,1-diazides, exploration of 1,1-diazides' properties is severely restricted. We believed that 1,1-diazides are just like a newborn baby, full of many new possibilities. Our work is a good example of exploring new methods for synthesizing such compounds, and provides a basis for researchers to explore the reactivities and properties of these molecules in future.

In fact, our studies about the further transformations of the products reveal new reactivities of 1,1-diazides and bring new insight into synthesis. For example, in traditional nucleophilic substitution reaction, the leaving groups of the substrates are commonly based on heteroatom-centered groups, such as halides or sulfonates. In contrast, we disclosed that carbon-center-based geminal diazidomethyl moiety (“CH(N₃)₂”) could serve as a formal leaving group, and its reactivity could be turned on by using simple “key” Lewis acid in our work. This fact brings new insight into nucleophilic substitution reaction. Our studies uncover new reactivity pattern with this class of compounds, and possess potential value in synthesis.

(iv) The reviewer pointed out that our work might be a relatively niche application of selenium catalysis whose impact on synthetic chemistry is likely to be minor.

Thank the reviewer for this comment. Unlike other popular research directions, selenium catalysis is a relatively niche research areas, but has received increasing attention in the past decade. Researchers hope to dig out the potentials of selenium catalysis to solve the synthetic issues that cannot be solved by other methods. As a matter of fact, selenium catalysis has become a powerful tool for the synthesis, and can be used to solve the synthetic issues that cannot be solved by other methods. Like this work, selenium catalysis enables the 1,1-diazides of alkenes. In the past decade, we have been devoted to the field of selenium catalysis to meet diverse synthetic world. We believe that this work could attract more attention from researchers, and is possible to trigger more useful transformations in synthetic chemistry.

3. Response to Reviewer 3’s comments:

The reviewer pointed out “*Zhao and coworkers developed an interesting method that provides a comprehensive exploration of catalytic 1,1-diazidation of alkenes, shedding light on a barely explored but valuable aspect of alkene functionalization. Significant contributions to the field of organic chemistry, specifically in the development of producing geminal diazides. Because this is an important area of research due to the potential applications of azides in synthetic chemistry. The authors have demonstrated the catalytic 1,1-diazidation of alkenes with well-defined conditions and have provided detailed information about the scale-up synthesis. Moreover, this study's thorough investigation of substrate scope, synthetic applications, and mechanistic insights significantly advances the field. The inclusion of crystallographic data for structures 2e adds credibility to the findings. Overall, the work presents novel findings in the catalytic 1,1-diazidation of alkenes and the synthetic applications of geminal diazides, making it a valuable contribution to the field of organic chemistry. Furthermore, the manuscript has been well-prepared and clearly presented, showcasing the expertise of the authors in this field. The supporting information is also well prepared and easy to follow. However, it would be beneficial*

for the authors to provide additional details regarding the substrate scope, reaction mechanism. Addressing these points would enhance the manuscript's scientific impact and provide a more comprehensive understanding of the reported research. Recommendation: Publication in Nature Communications is supported after Major revision.” Thanks for this positive comment. The reviewer raised some questions and gave some suggestions as follows:

- (1) The reviewer pointed out “1) As reported by Wolloch et al. (ANNALEN DER CHEMIE-JUSTUS LIEBIG, 1975, 2339). 1,3-dienes can undergo group migration to yield diazides albeit with low yield. This reviewer wonders what are the biggest difference and challenge when compared to Wolloch’s work and this manuscript.”

Thank the reviewer for pointing out. We tried our best to get the mentioned paper (Wolloch et al., *Annalen Der Chemie-Justus Liebig*, 1975, 2339), no matter via searching in Scifinder or directly accessing the official website of the journal. Unfortunately, we did not find the paper. Neither name (Wolloch) nor year (1975) or page number (2339) was matched in the journal *Annalen Der Chemie-Justus Liebig*. For this reason, we did not know what exact transformation was involved in that paper. Therefore, it is difficult for us to comment the difference between that work and ours.

- (2) The reviewer pointed out “2) For alkylated substrates, such as *1at-aw*, the beta-H elimination often happened to form alkenylated azides. Are there any such side products observed for reactions with *1at-aw*? If not, the excellency in selectivity should be discussed?”

Thank the reviewer very much for pointing out. Indeed, we did not observe alkenylated azide side products generated from beta-H elimination for the reactions of alkylated substrates **1at-aw**. However, if there was an electron-withdrawing group on the allylic position, the beta-H elimination could occur. For example, allylic azide was observed as the main product when allylic nitrile was utilized as substrate (see section 4.4 Limitation of the Developed Method in Supplementary Information). For the common alkylated substrates, the excellent migratory selectivity outcompeting the selectivity of beta-H elimination might be ascribed to relatively lower acidity of β -hydrogen on the corresponding selenenylated intermediates.

To make this point clearer, we added the following description in the manuscript: “Interestingly, for the common alkylated substrates **1at-1aw**, alkenylated azides generated from possible side reaction via β -H elimination were not observed. This excellent selectivity of migration versus β -H elimination might be ascribed to relatively lower acidity of β -hydrogen on the corresponding selenenylated intermediates. If there was an electron-withdrawing group on the allylic position of the substrates, β -H elimination could occur rather than migratory functionalization (see the Supplementary Information)^{54,56}.”

(3) The reviewer pointed out “3) *The substrate scope is somehow limited to arylated alkenes. Can 1,3-dienes be used as the substrates or other substrates tolerated?*”

Thank the reviewer for this comment. When different 1,3-dienes were utilized as substrates in our catalytic system, the reactions were messy. Using different 1,3-dienes, we could detect the desired 1,2-diazides sometimes, but in low yields. We also tried to use enynes as substrates. The reactions did not occur or gave the mixed products. The results are summarized as below (also collected in the section 4.4 Limitation of the Developed Method in the Supplementary Information):

On the basis of our studies, arylated alkenes are suitable for our reaction condition to produce the desired products in high yields with good selectivities. For other substrates, new reaction conditions might be needed to give the desired products in high yields.

(4) The reviewer pointed out “4) *PhSeSePh is a common catalyst for many reactions. In some of them, the Se(IV) species was supposed to be the reactive catalysts, for example, in Denmark’s paper, Nature Chemistry, 2015, 7, 146. In this paper, PhSeBF₄ was regarded as the real catalyst. Could the authors provide more explanation on why PhSeBF₄ was the active catalyst?*”

Thank the reviewer for this question. What the reviewer concerned about might be whether Se(IV) species or Se(II) species initiates the transformation by interaction with alkene in the reaction using PhSeSePh as catalyst precursor. Although there is no solid consensus on this issue in the field of selenium redox catalysis, the difference between these two ways is actually not significant. Both ways go through the same Se(IV) intermediate before deselenenylation (for the Se(II) species initiated process, the Se(II) intermediate is oxidized to Se(IV) intermediate before deselenenylation). Besides, both ways generate the Se(II) species after deselenenylation (for the Se(IV) species initiated process, the Se(II) species is oxidized to Se(IV) species before interacting with alkene).

In Denmark's work (*Nat. Chem.* **7**, 146–152 (2015)), although the authors proposed the Se(IV) species as the reactive catalyst, they could not exclude a Se(II) species initiated process either. In that paper, they mentioned “To complete the cycle, oxidation of PhSeCl by **11** (oxidant in that paper) in the presence of chloride ions could regenerate the PhSeCl₃ catalyst, although it is possible that addition of PhSeCl to the alkene could precede the oxidation of Se(II) to Se(IV)”. In fact, in the Denmark's subsequent work (*Tetrahedron* **75**, 4086-4098 (2019)), they considered that both processes are possible and indistinguishable. Besides, in their following similar works (Ref. 48: *J. Am. Chem. Soc.* **141**, 19161–19170 (2019); Ref. 49: *J. Am. Chem. Soc.* **143**, 13408–13417 (2021)), they proposed that the reactions go through a Se(II) species initiated process instead of the Se(IV) species initiated one.

According to the literature (*Inorg. Chem.* **47**, 4712–4722 (2008)), organoselenium(IV) triazides are very unstable even at -50 °C, and will decompose immediately when being slowly warmed up. In contrast, our reactions were run at room temperature, which does not benefit the existence of selenium(IV) triazides. On the other hand, selenenylated azide intermediate **7** was observed according to our experiment (Fig 3f), suggesting that the Se(II) species initiated process seems to be more feasible.

Based on the considerations above, although we could not rule out that our reaction was initiated by Se(IV) species, we are inclined to believe that our system is in a Se(II) species initiated process. To help readership understand this point better, we added the following description in the mechanistic part of the manuscript: “Although we cannot rule out that our reaction goes through a selenium(IV) species initiated process⁷⁶, we are inclined to believe that a selenium(II) species initiates the reaction at the beginning since temperature-sensitive selenium(IV) polyazides might be difficult to exist under the reaction temperature⁷⁷.” New references 76 (Cresswell, A. J., Eey, S. T. -C., & Denmark, S. E. Catalytic, stereospecific syn-dichlorination of alkenes. *Nat. Chem.* **7**, 146–152 (2015)) and 77 (Klapötke, T. M., Krumm, B., & Scherr, M. Studies on the properties of organoselenium(IV) fluorides and azides, *Inorg. Chem.* **47**, 4712–4722 (2008)) were added, and the order of subsequent references was changed accordingly.

(5) The reviewer pointed out “5) In Fig.4 nucleophilic attack of species (I) by TMSN₃ to form species (II) was proposed. However, this reviewer doubt that the carbon attached with PCP should be more electrophilic. Is this correct?”

Thank the reviewer very much for this good question. Actually, when the seleniranium ion part connects to the electron-rich *p*-methoxyphenyl (PMP) group and the electron-deficient *p*-cyanophenyl (PCP) group, the carbon reactive site attached by the PMP group is more electrophilic than the site attached by the PCP group. It is because the stabilization of the partial

positive charge by the electron-rich PMP group is more than that by the electron-deficient PCP group. Please see the scheme below. So the reaction pathway is as drawn in the manuscript. Similar phenomenon of regioselectivity was observed, and there are some related discussions in the literature (Ref. 49: *J. Am. Chem. Soc.* **143**, 13408–13417 (2021)).

To make this point clearer, we added the following description in the manuscript: “It is noted that the attack of the azide group toward intermediate I is in excellent regioselectivity because the reactive site attached by the p-methoxyphenyl group is more electrophilic than the site attached by the p-cyanophenyl group⁴⁹.”

(6) The reviewer pointed out “6) Migratory 1,1-difunctionalization of alkenes is an interesting strategy for the synthesis of geminal functionalized compounds. Instead of discussion 1,1-difluorination, key literatures for 1,1-diborylalkanes from terminal alkenes should also be mentioned and cited.”

Thank the reviewer for this suggestion. We added the relevant description in the introduction of the manuscript: “It is worth mentioning that although elegant 1,1-difluorination and 1,1-diborylation of alkenes has been achieved by using iodine catalysts and metal catalysts, respectively^{37-42,51-53}, the azidative version has not be realized until our work.” New references 51-53 (51: Li, L., Gong, T., Lu, X., Xiao, B., & Fu, Y. Nickel-catalyzed synthesis of 1,1-diborylalkanes from terminal alkenes. *Nat. Commun.* **8**, 345 (2017). 52: Teo, W. J. & Ge, S. Cobalt-catalyzed diborylation of 1,1-disubstituted vinylarenes: a practical access to branched gem-bis(boryl)alkanes. *Angew. Chem. Int. Ed.* **57**, 1654–1658 (2018). 53: Wang, X. et al. Zirconium-catalyzed atom-economical synthesis of 1,1-diborylalkanes from terminal and internal alkenes. *Angew. Chem. Int. Ed.* **59**, 13608–13612 (2020)) about 1,1-diborylation of alkenes are cited, and the order of subsequent references was changed accordingly.

REVIEWERS' COMMENTS

Reviewer #3 (Remarks to the Author):

Zhao et al. have revised the manuscript by adding more examples and providing more comments for better understandings. As replied, all the concerns have been carefully addressed, except for question one. This reviewer agrees that the method is a new one for catalytic 1,1-diazidation of alkenes. Therefore, it may be suitable for the acceptance of Nature Communications journal.

Thank reviewer 3 very much for the comment to help us to improve the quality of our manuscript. In the following is our point-by-point response to this comment.

1. Response to Reviewer 3's comments:

The reviewer pointed out “*Zhao et al. have revised the manuscript by adding more examples and providing more comments for better understandings. As replied, all the concerns have been carefully addressed, except for question one. This reviewer agrees that the method is a new one for catalytic 1,1-diazidation of alkenes. Therefore, it may be suitable for the acceptance of Nature Communications journal.*”

Reviewer 3's original question one is “1) As reported by Wolloch et al. (ANNALENDER CHEMIE-JUSTUS LIEBIG, 1975, 2339). 1,3-dienes can undergo group migration to yield diazides albeit with low yield. This reviewer wonders what are the biggest difference and challenge when compared to Wolloch's work and this manuscript”. We tried our best to get the mentioned paper (Wolloch et al., *Annalen Der Chemie-Justus Liebig*, 1975, 2339), no matter via searching with Scifinder or directly accessing the official website of the journal. Unfortunately, we did not find the paper. We did not know what exact transformation was involved in that aged paper. So, it is difficult for us to comment the difference between that work and ours.